# Development of an Automated Online Flow Cytometry Method to Quantify Cell Density and Fingerprint Bacterial Communities

**DOI:** 10.3390/cells12121559

**Published:** 2023-06-06

**Authors:** Juan López-Gálvez, Konstanze Schiessl, Michael D. Besmer, Carmen Bruckmann, Hauke Harms, Susann Müller

**Affiliations:** 1Department of Environmental Microbiology, Helmholtz-Centre for Environmental Research, Permoserstraße 15, D-04318 Leipzig, Germany; juan.lopez-galvez@ufz.de (J.L.-G.); carmen.bruckmann@ufz.de (C.B.); hauke.harms@ufz.de (H.H.); 2onCyt Microbiology AG, Marchwartstrasse 61, 8038 Zürich, Switzerland; konstanze.schiessl@oncyt.com (K.S.); michael.besmer@oncyt.com (M.D.B.)

**Keywords:** automated online flow cytometry, fingerprinting, cell density determination, pattern analysis, process control

## Abstract

Cell density is an important factor in all microbiome research, where interactions are of interest. It is also the most important parameter for the operation and control of most biotechnological processes. In the past, cell density determination was often performed offline and manually, resulting in a delay between sampling and immediate data processing, preventing quick action. While there are now some online methods for rapid and automated cell density determination, they are unable to distinguish between the different cell types in bacterial communities. To address this gap, an online automated flow cytometry procedure is proposed for real-time high-resolution analysis of bacterial communities. On the one hand, it allows for the online automated calculation of cell concentrations and, on the other, for the differentiation between different cell subsets of a bacterial community. To achieve this, the OC-300 automation device (onCyt Microbiology, Zürich, Switzerland) was coupled with the flow cytometer CytoFLEX (Beckman Coulter, Brea, USA). The OC-300 performs the automatic sampling, dilution, fixation and 4′,6-diamidino-2-phenylindole (DAPI) staining of a bacterial sample before sending it to the CytoFLEX for measurement. It is demonstrated that this method can reproducibly measure both cell density and fingerprint-like patterns of bacterial communities, generating suitable data for powerful automated data analysis and interpretation pipelines. In particular, the automated, high-resolution partitioning of clustered data into cell subsets opens up the possibility of correlation analysis to identify the operational or abiotic/biotic causes of community disturbances or state changes, which can influence the interaction potential of organisms in microbiomes or even affect the performance of individual organisms.

## 1. Introduction

The microbiome is an integral part of the human body and lifestyle and influences health and well-being in countless ways [1]. The use of microbial consortia has recently also gained importance for the biotechnological production of high-value compounds [2,3,4]. This is because consortia can use a wider range of carbon sources than individual organisms, catalyze longer and more complex biosynthetic pathways, and degrade inhibitory by-products. In order to grow, multiply and perform a desired task or produce a desired product, some individual bacteria in communities may require environments that are different from others and that can be supported by certain operating conditions [5]. The precise tracking of cell dynamics within microbial communities, therefore, makes it possible to identify functionally active subgroups of cells over long periods of time using correlation analysis. This opens up the possibility of manipulating microbial communities through different process controls to support and stabilize specific cell types in their desired functionalities.

Cell density is the most commonly used parameter in processes of industrial biotechnology, along with substrate and product analyses, providing information on process statuses and enabling process control [6,7]. It is the key parameter for adjusting all key process control variables, such as the feed rate, harvest time, product recovery, etc. Since cells (biomass) are the active transformers in any process, the level of their abundance is highly relevant. However, traditionally and through manual sample preparation, cells are analyzed as a bulk parameter per volume, such as dry or wet weight. Such manual handling is imprecise, prone to human error and time-consuming. The delay between sampling and obtaining biomass information is substantial and hinders meaningful and fast decision-making [7,8]. Therefore, several online methods have been developed for continuous monitoring in real-time. However, all of these are based on either spectroscopic or capacitance sensors and report the cell’s biomass only as a bulk parameter [7,9,10,11]. Their inability to discriminate between different species can severely limit applications where influencing the interaction between certain cell types helps catalyze significant metabolic pathways [12]. To distinguish between cell types, 16S rRNA gene amplicon sequencing is often applied [13,14]. While this adds the necessary organismic information, it can only be performed discontinuously, manually and offline. In addition, there is a delay between sampling and immediate data processing due to the lengthy evaluation procedures required.

To overcome these limitations, techniques that can identify individual subcommunities by differentiating between the cell types are needed in real-time. For mammalian and yeast cells, an optical sensor was developed that allows online microscopical monitoring [15,16]. For bacterial cells, such techniques are not yet available, as the differentiation is currently based on monitoring either the morphological changes of cell types or the changes in autofluorescence intensities below 300 nm, e.g., based on NAD/NADP redox ratio analysis [17,18]. These properties cannot be obtained from single bacterial cells, especially when monitoring complex bacterial communities, since the morphological and autofluorescent parameters are far too similar and variable to be associated with specific individual cell types.

As an offline tool, conventional flow cytometry has been successfully used in biotechnological processes to measure not just the variations in overall cell concentration but also to provide information on the different cell types within productive communities [19] in wastewater [20] or in anaerobic digesters [21], as well as for the control of *E. coli* and *S. cerevisiae* cultures in bioreactors [22] or for the fingerprinting of different *Lactobacillus* strains [23]. Beyond biotechnology, the offline flow cytometry of microbial communities’ states has a wide range of applications, e.g., in drinking water [24,25], in rat and mouse microbiomes [26,27] or for electricity generation [28,29].

Following the proportions of cell types online is desirable since it allows for the targeted and in-time manipulation of a community’s composition by changing the control parameters. Fully automated online flow cytometry is already applied to drinking water systems for measuring cell concentrations [30,31] or live/dead cell proportions [32]. A rough differentiation of drinking water communities was attempted by calculating high nucleic acid/low nucleic acid (HNA/LNA) ratios [33,34]. Online flow cytometry was also already used to measure pure cultures, e.g., lipid accumulation and the cell growth of yeast [35]. However, an online analysis procedure of complex microbial communities that can distinguish a large number of subcommunities in microbial bioprocesses with typically high cell densities is not yet feasible.

Here, we propose an automated monitoring system comprising hardware, software and an automated workflow to monitor both the absolute cell abundance and community composition online with a high temporal resolution in dense samples that are typical for biotechnology. This should make it possible to create and evaluate community fingerprints, i.e., profiles of the community structure, at any time. To this end, we combined (i) the existing automation technology for sampling and sample preparation with (ii) microbial community flow cytometry and (iii) proven staining protocols for subcommunity differentiation. In order to achieve the performance required to meet all the above objectives, each of these components had to be further optimized, simplified and standardized before being combined.

To this end, several automation steps were necessary: First, the cell density must be reduced to a level that can be measured by a customized flow cytometer. Second, the samples had to be treated to resolve t the community into different subsets of cells in order to create a fingerprint. This requires the cells to be automatically fixed and stained, which is adapted from a protocol by Günther et al. (2008) in which NaN_3_ fixation was already used [36]. Third, we also demonstrate that the obtained single-cell data can be automatically evaluated [37] to interpret the dynamics of subsets of cells of a community, which is a prerequisite for directly testing the dependencies of subsets using the process control parameters. This would provide the operator with the necessary information to change the process parameters within available control periods.

## 2. Materials and Methods

### 2.1. Cultivation and Harvest of Bacterial Cells

The pure strains (i) *Kocuria rhizophila* DSM 348 (German Collection of Microorganisms and Cell Cultures (DSMZ; Leibniz Institute, Braunschweig, Germany), (ii) *Paenibacillus polymyxa* DSM 36 (DSMZ; Leibniz Institute Braunschweig, Germany) and (iii) *Stenotrophomonas rhizophila* DSM 14405 (DSMZ; Leibniz Institute Braunschweig, Germany) were first cultivated on LB agar plates (Lysogeny broth, 30 °C, 72 h). Twenty mL of liquid LB medium was then inoculated with one colony of the respective strain and cultivated (30 °C, 24 h, 150 rpm). After 24 h, the OD_700 nm = 0.5 cm_ (Ultrospec 1100 Pro, Amersham Biosciences, Amersham, UK) of the preculture was measured, and the necessary volume to inoculate the main culture at an initial optical density of OD_700 nm = 0.5 cm_ = 0.05 was calculated (30 °C, 24 h, 150 rpm) and used to inoculate 100 mL of LB medium. Then, the cells were harvested by centrifugation (3200× *g*, 4 °C, 10 min; 5810 R centrifuge, Eppendorf SE, Hamburg, Germany) and adjusted to OD_700 nm = 0.5 cm_ = 1 in PBS buffer (Appendix A). For the bacterial samples comprising all three strains, each strain was first cultivated and harvested separately, as described above, and then mixed at a relative OD ratio of 80:1:19 for *P. polymyxa*, *S. rhizophila* and *K. rhizophila*, respectively, in line with Cichocki et al. (2020) [38]. The term mock community (MC) is used for the artificial assembly of these three strains.

### 2.2. NaCl/NaN_3_/EtOH Fixation

NaCl/NaN_3_/EtOH fixation was performed both manually (by a human operator) and online (by the OC-300 CytoFLEX instrumentation). The stabilized cells were first diluted in PBS buffer at a ratio of 1:1. Then, the respective amounts of NaCl (30% (*m*/*v*) stock solution (Merck KGaA, Darmstadt, Germany), NaN_3_ (20% (*m*/*v*) stock solution (Merck KGaA, Darmstadt, Germany) and EtOH (70% (*v*/*v*) stock solution (Chemsolute, Renningen, Germany) were calculated to obtain the desired concentrations. NaN_3_ and EtOH were tested at concentrations between 1 and 10%. NaCl was tested at concentrations between 1 and 20%. Different incubation times, ranging from 5 to 30 min, were also tested. The best combination that allowed a good resolution of the different strains of the MC was 1% NaN_3_, 20% NaCl and 10% EtOH at an incubation time of 10 min.

### 2.3. Paraformaldehyde (PFA)/EtOH Fixation for Standard Cytometric Analysis

PFA/EtOH fixation was performed as a sample pre-treatment for the offline experiments, as described by Cichocki et al. (2020) [38]. In short, cells were adjusted to an OD_700 nm = 0.5 cm_ of 0.5 (~10^9^–5 × 10^9^ cells), centrifuged at 3200× *g* at 4 °C for 10 min and incubated for 30 min in 2 mL of a 2% PFA solution (Appendix A) at room temperature (RT). After this, the PFA was discarded by centrifugation (3200× *g*, 4 °C, 10 min), and finally, the cells were resuspended in the same volume of a 70% EtOH fixation solution for at least 1 h at RT before analysis. Cells can be stored at −20 °C for up to 6 months.

### 2.4. DAPI Staining and Final Dilution for Standard Cytometric Analysis

The fixed cells were washed with PBS buffer by centrifugation (3200× *g*, 4 °C, 10 min) and adjusted to an OD_700 nm = 0.5 cm_ = 0.25. Then, the cells suspended in PBS buffer were mixed with a 1 µM DAPI staining stock solution (Appendix A) at a 1:2 ratio and incubated for 10 min in the dark at RT. Thereafter, the stained cells were diluted in double distilled water at a 1:20 ratio and immediately measured.

### 2.5. Flow Cytometer

A CytoFLEX S flow cytometer (Beckman Coulter, Brea, CA, USA), operated with the CytExpert software (Beckman Coulter, Brea, CA, USA), was used. The machine is equipped with 375 nm (60 mW), 488 nm (50 mW) and 638 nm (50 mW) lasers. The 488 nm laser light was used for the detection of forward scatter (FSC) (488/8 nm band-pass) and side scatter (SSC) (488/8 nm band-pass, trigger signal). The DAPI fluorescence (450/45 nm band-pass) was measured using the 375 nm laser for excitation. The fluidic system ran at a constant speed of 60 µL/min. For the optical calibration of the device in the logarithmic range, 0.5 µm and 1.0 μm UV fluoresbrite microspheres (Polysciences, Cat. No. are 18339 and 17458, Warrington, PA, USA, Appendix A) were used.

### 2.6. OC-300

The OC-300 Duo automation module (onCyt Microbiology, Zürich, Switzerland) is specifically designed for use in the automated flow cytometry of bacterial communities in technical and environmental water ecosystems, such as fresh water, drinking water and wastewater [32,39,40,41]. In this study, we used it to count and differentiate the cell types in bacterial communities using DAPI staining. Data were recorded by the cyOn control software (onCyt Microbiology, Zürich, Switzerland), which operated both the OC-300 and the connected CytoFLEX during the online experiments. Two valves, each with twelve different ports, allowed the intake and dilution of a sample and the intake of the reagents for fixation and staining via a nondisposable syringe. They also provided the connection between the incubation chambers (Figure 1). In the three incubation chambers, the different solutions were mixed with air bubbles and the resulting sample was then placed in the syringe, which had been rinsed with water in the meantime, from where it was transferred to the CytoFLEX (Figure 2).

### 2.7. Online Flow Cytometry Workflow for the Dilution, Fixation and Staining of Bacterial Cells

The sample that the OC-300 draws from was incubated in an Erlenmeyer flask at RT and stirred at 300 rpm. The workflow implemented here for the OC-300 is the following (Figure 2): the OC-300 first dilutes the sample 1:1 with PBS buffer in chamber 1 (C1). Then, the diluted sample is moved to chamber 2 (C2) along with NaCl 30%, NaN_3_ 20% and EtOH 70% at a ratio of 1:4.75:0.35:1 (sample: NaCl: NaN_3_: EtOH), where the sample is incubated for 10 min for fixation. Then, the fixed sample is diluted at a ratio of 1:2 with 1 µM DAPI staining solution, and the mix is incubated for 10 min at RT in C3 for staining. The stained sample is finally diluted at a ratio of 1:20 in MilliQ H_2_O, prepared by the MilliQ IQ 7000 Ultrapure Lab Water System (Merck KGaA, Darmstadt, Germany). Then, the stained sample is sent to the CytoFLEX to be measured for 1 min. After every measurement, the device cleans itself and prepares for the next sampling. Considering the fixation and staining times, as well as the cleaning of the device between samples, a new sampling and measurement is made every 45 min.

### 2.8. Cell Count Measurement

Cell numbers were measured by using the method developed for the automated analysis of diluted, fixated, DAPI-stained bacterial cells (see Section 2.7).

### 2.9. Stability of Measured Cell Concentration over Time Using the Online Flow Cytometry Workflow

Either the live cells in PBS or the manually PFA/EtOH pre-fixed cells were used as the input cells for the online flow cytometry workflow. The online workflow then sampled and measured the cell concentration every 45 min, as described above. The flow cytometry data were then processed by the analysis software cyPlot (onCyt Microbiology, Zürich, Switzerland). In this software, gates were set for each bacterial population, which was used to determine the concentration of each bacterial strain.

### 2.10. Self-Cleaning Capabilities of the OC-300 Automation Unit between Measurements

It is necessary to verify the proper self-cleaning of the automation device between measurements during the online flow cytometry workflow. When the continuous measurement of a bacterial community is performed, bacteria from the previous sample should not significantly contribute to the next measured sample (low sample carryover). The cleaning of the automation device is performed by running a cleaning solution through all the ports, chambers and syringes after each measurement. After the cleaning solution is run through the device, two steps of rinsing with MilliQ H_2_O are passed through the device to eliminate all the droplets of the cleaning solution left behind. Four different cleaning solutions were tested: 100% FlowClean cleaning agent (No: A64669, Beckman Coulter, Brea, CA, USA), 10% FlowClean cleaning agent (diluted in MilliQ H_2_O), 1% FlowClean cleaning agent (diluted in MilliQ H_2_O) and MilliQ H_2_O. To measure the efficiency of each cleaning solution, the cell concentration in two samples was measured alternately by the online flow cytometry workflow for at least 24 h. The first sample was a bacterial sample comprising live cells from one of the three strains of the MC in PBS buffer adjusted to an OD_700 nm = 0.5 cm_ = 1, and the second sample was MilliQ H_2_O. After the cleaning solution ran through the device, two runs of MilliQ H_2_O were interposed to eliminate any cleaning solution left in the device. After the procedure, the next sample was measured.

### 2.11. Bioinformatics Evaluation

The cell concentration and relative abundance of each strain were calculated using the software cyPlot. In this software, gates are manually set, encompassing cells with similar characteristics that cluster together as a subpopulation. Flow cytometric 2D plots were also created with the software FlowJo (BD Biosciences, Franklin Lakes, NJ, USA). In addition, this software was also used for the analysis of subpopulations of *P. polymyxa*, *S. rhizophila* and *K. rhizophila* (C1, C2 and Cx, respectively), which allows the setting of gates as ellipses, a prerequisite for subsequent automated evaluation procedures. The resulting gate template was used for the cell-number-related evaluations in this study. The nonmetric multidimensional scaling (NMDS) comparison was performed using the flowCyBar software [43]. Furthermore, the cell numbers per gate were also automatically provided by using the bioinformatic tool, flowEMMi, v2, which was included in the biTCa Analyze Tool graphical user interface (GUI) developed by Bruckmann et al. (2022) [37]. The parameters used for the automatic gating of 132 samples and the automatic generation of the gate template were alpha = 0.7, min minor = 1, convergence = 0.01, max cluster = 5, min cluster = 3 and cluster bracket = 3. Before flowEMMi v2 was applied, the instrumental noise was virtually removed from the 2D plots.

## 3. Results

### 3.1. Automated Dilution, Fixation, Staining and Analysis of Microbial Community Samples

We have developed a new automated online workflow for the cell quantification of bacterial communities that allows for accurate discrimination between different subcommunities. The new online method does not require any centrifugation steps to separate the cells from actual treatment solutions, which cannot be included in the online procedures. To ensure the reliable establishment of the new online measurement method, a bacterial mock community (MC) [38] was used as a biological standard. This MC comprised three different bacterial strains: *Kocuria rhizophila*, *Stenotrophomonas rhizophila* and *Paenibacillus polymyxa*. These strains were either Gram-negative (*S. rhizophila*) or Gram-positive strains (*K. rhizophila* and *P. polymyxa*) and differed in their cell wall composition, cell size, GC content and chromosome size [38]. In combination, and thus as an artificial mixed community, they are ideally suited for testing and verifying new methods.

The first step in developing an automated online procedure was to develop a dilution, fixation and staining method compatible with the capabilities of the OC-300. In the next step, the MC was used to check whether the strains could be distinguished on the basis of cell size and DNA content per cell by DAPI staining, even in subpopulations with different chromosome copy numbers. In addition, it was tested if the MC could be separated from background noise. Thirdly, fingerprinting data should be obtained in less than 1 h, allowing for continuous analysis of dynamic communities.

Even though the separation between the three different strains of the MC was not as clear as the one achieved by the standard method developed by Cichocki et al. (2020) [38], which performs a sequential PFA/EtOH fixation procedure for 2 h that involves several washing and centrifugation steps and overnight staining with a two-step 0.24 µM DAPI staining solution (Appendix A), the new online procedure allows us to fully discriminate the strains of the MC. In addition, the *P. polymyxa* and *S. rhizophila* physiological sub-states were successfully differentiated from all other components of the MC (Appendix A). Overall, with this new dilution, fixation and staining method, the subpopulations of a bacterial community can now be determined within only 45 min.

In the next step, this new protocol was implemented as an automated online method by programming the different sequential steps described in Materials and Methods. In Figure 3, we demonstrate that the community fingerprints are highly comparable between a fully automated online and manual workflow of the sample preparation and measurement for all three strains, separately as well as for the combined MC.

More specifically, when only the pure *K. rhizophila* strain was measured (Figure 3a,e), the obtained population pattern was the same, irrespective of whether sample preparations and measurements were performed online or manually. This also holds true for the samples comprising pure *S. rhizophila* (Figure 3b,f) and pure *P. polymyxa* (Figure 3c,g), each of them showing two different subpopulations occupying the same respective positions in the 2D plots. When all strains were mixed to assemble the MC (Figure 3d,h), all cell populations and their subpopulations maintained their positions unchanged. The consistent fingerprint-like pattern showed that the different strains of the MC and their subpopulations could be distinguished when dilution, fixation and staining were performed in an automated manner. The measurement of fingerprint-like patterns for microbial communities apart from instrumental noise was possible within a time period of 45 min.

### 3.2. Reliable Self-Cleaning Procedures of the OC-300

It is essential to verify sufficient self-cleaning of the OC-300 automation device between measurements to ensure that the carryover between consecutive measurements is low. After each measurement, the device was cleaned with an automated workflow, including the chambers, relevant tubing connection, the syringe, valves and nozzle of the flow cytometer (see materials and methods). Four different cleaning solutions were tested: MilliQ H_2_O, the undiluted 100% FlowClean cleaning agent, as well as the 10% and 1% FlowClean cleaning agent solutions. To test the efficiency of each cleaning solution, two samples were measured alternately using the online flow cytometry workflow, as described in Section 2.10. The percentage of carryover from the sample to the negative control (MilliQ H_2_O) was calculated for each cleaning solution and each strain as a pure culture. The results of these experiments are shown in Table 1.

Table 1 shows that both the 10% and the 1% FlowClean cleaning agent solutions had the best cleaning performance of the four cleaning solutions. These solutions reduced the carryover to 3.1% and 3.8%, respectively, when the bacterial sample comprised *P. polymyxa*, compared to cleaning with MilliQ H_2_O alone with a carryover of 7.6%. In the case of *S. rhizophila*, the 10% and 1% FlowClean cleaning agent solutions had a carryover of 2.8% and 2.9%, respectively. For *K. rhizophila*, the 10% and 1% FlowClean cleaning agent solutions reduced the carryover to 2.1% and 1.1% from 3.1%, compared to MilliQ H_2_O. However, even though the 10% and 1% FlowClean cleaning agent solutions have the same cleaning efficiency, the use of the 10% solution slightly altered the bacterial patterns obtained by the automated device (Appendix A). This effect was also seen when using the pure FlowClean cleaning agent solution (Appendix A). For this reason, the 1% FlowClean cleaning agent was chosen as the cleaning solution for all further experiments.

### 3.3. Reliability of Automatically Determined Cell Counts

The automated online measurement of cell concentrations is another advantage of the OC-300 CytoFLEX combination because it provides immediate information on the variations in cell density. For this measurement, the newly developed DAPI method was used, in which cells were not lost through the cell preparation procedure. This clearly separated the cells from the instrumental noise, which did not affect the cell counts. By measuring the same cell concentration over a period of 48 h, we wanted to demonstrate the low random error and high technical reproducibility of the online method. First, live cells of the MC stirred in 50 mL of PBS were used as an input sample, where each of the strains had the same OD concentration. Unexpectedly, the data showed a considerable variation in cell concentrations over time for all three strains of the MC, with a variation coefficient of about 24%, 27% and 28% for *K. rhizophila*, *P. polymyxa* and *S. rhizophila*, respectively (Appendix A). This was surprising because the cell processing procedure of the OC-300 contains a fixation step, and a cell number variation due to the biological processes, such as cell division, was not expected. However, before entering the OC-300, the live cells were maintained in stirred PBS buffer for 48 h. Therefore, we assumed that the variation in cell number was not introduced by the device.

The following experiments were performed with pre-fixed cells instead of live cells to avoid the biological processes already taking place in the stirred PBS-buffered feed flasks. The cells were pre-fixed manually with PFA/EtOH, as described by Cichocki et al. (2020) [38], in order to assure that the biology features of the bacterial sample would remain constant over a time period of at least 34 h. This experiment was replicated three times (gate template Appendix A). Figure 4a shows the normalized cell concentration average for three biological replicate experiments. The cell concentration of each experiment was normalized against its own average. The coefficient of variation for each strain was 15%, 14% and 15% for *K. rhizophila*, *P. polymyxa* and *S. rhizophila*, respectively. Notably, the cell concentration measurements were found to be still highly variable during the first five sampling points from 0 h to 3 h. If these time points are excluded from the variation of coefficient calculation, the values are reduced to 14%, 9% and 13% for *K. rhizophila*, *P. polymyxa* and *S. rhizophila*. These data show that the automated cell concentration measurements remained much more stable when the fixed cells were used as the input sample (Figure 4a) compared to when the same experiment was conducted with live cells (Appendix A).

This is corroborated by the NMDS plots, shown in Figure 5. The data produced by the cell proportions obtained from the experiment with live cells are clearly different from those obtained from the experiments with pre-fixed cells. The NMDS plots from the experiments with pre-fixed cells converge more strongly, suggesting that the large fluctuations in cell concentration measurements, which were observed in Appendix A, were not caused by the cell processing procedure of the OC-300 but rather by the biological changes in the bacteria that occurred in the stirred feed tank during continuous measurements for at least 34 h.

However, especially for the pre-fixed *P. polymyxa* measurements, we found an increasing trend in the cell number over time (Figure 4b,d,f). In parallel, the forward scatter of the *P. polymyxa* was decreasing. The other parameter in the 2D plots is the DAPI fluorescence, which provides information on the number of chromosomes per cell. For *P. polymyxa,* two subpopulations were found, with the C1 subpopulation comprising single cells with one chromosome content, whereas the C2 subpopulation contained cells with double chromosome content. The DAPI fluorescence intensity of the C1 subpopulation was 5–6 × 10^4^ relative DAPI intensity units, whereas, for the C2 subpopulation, the value was 1 × 10^5^ relative DAPI intensity units, thus supporting this assumption. The cells of subpopulation C2 may be cells whose higher numbers of chromosomes indicate that they may still be interconnected duplicate cells that have not yet undergone complete cell division. In order to assess whether the increase in cell concentration and the decrease in a forward scatter of pre-fixed *P. polymyxa* cells were related, the bioinformatics tool flowEMMi v2 [37], which was specifically developed for the automatic gating of cell subpopulations in 2D plots, was used. A gate template out of the 132 pre-fixed samples from Figure 4 was created by flowEMMi v2 (Figure 6c), which was used to calculate the number of cells in the subpopulations C1 and C2 of *P. polymyxa* in addition to the cell numbers of *S. rhizophila* and *K. rhizophila*. In *P. polymyxa*, it was found that the cell number in subpopulation C1 increased significantly, while the number in subpopulation C2 increased only slightly. This increase in cell number per subcommunity goes along with a decrease in forward scatter, thus confirming that stirring the fixed samples might cause breakage of doublet cells or smaller cell aggregates. This is even further confirmed by the use of a different but manual gating procedure (FlowJo, Appendix A), in which a third subpopulation of *P. polymyxa* (Cx, 2–3 × 10^5^ mean relative DAPI intensity units) clearly decreased in cell abundance over the 34 h of stirring (Figure 6d). In Figure 6d, we can see that at the start, the Cx subpopulation has a proportion of 0.10; however, as time passes by, this reduces until it disappears almost completely. Additionally, the subpopulation comprising single cells (C1) increases constantly over time, while the C2 subpopulation had an initial increase in cell number and then remained constant.

These data show, firstly, that the cell count measurement is not stable during the first 4 h of automated online analysis and, secondly, that the cell counts are, on average, reliably measured between biological samples, as shown by the normalized data set in Figure 4a. Thirdly, it demonstrates the high sensitivity of the method, which is able to detect small variations in the population numbers caused by physiological state changes in the subpopulations.

## 4. Discussion

A circular bioeconomy is one of the main components for sustainable development required in the near future. This type of economy is largely based on the use of microbial consortia for the biotransformation of waste into valuable products, most notably biofuels [44,45]. However, the use of microbial communities in biotechnological processes is limited because of their complexity when it comes to controlling community composition and its functionality [12]. Microbiomes and the complex interaction of their members with the host also make dense sampling campaigns in monitored environments a prerequisite for gaining a deeper understanding and finding solutions to disrupt or support them. The first step for gaining proper control of a bacterial community is, therefore, to obtain frequent and reliable information on its composition and active members. Rapid and reliable monitoring of a bacterial community—without significant time delay—enables a more precise and faster control of interventions, e.g., switching metabolic pathways to increase the productivity or interactions between individual cells or with the host.

Flow cytometry has already been used in the past to monitor bacterial communities to detect dynamic changes in their composition [27,46,47]. HNA/LNA ratios were used widely in flow cytometry in the past for the determination of two subsets of communities, taking up either low or high amounts of SYBR Green [34,48]. However, this division of bacteria into only two subpopulations provides an incomplete picture of how a community is composed and how it changes.

A higher resolution allows the partitioning of communities into more subgroups to obtain the fingerprints of current states of bacterial consortia, which enables the calculation of the ecological parameters of these communities [49]. DAPI as a stain for generating community fingerprints has never been used in conjunction with online flow cytometry thus far. DAPI was chosen over SYBR Green due to its better resolution of microbial communities into cell subsets and, thus, its higher discriminative power when dealing with complex microbial communities [38,49]. Given that we wanted to use online flow cytometry to assess the community dynamics using a fingerprinting approach based on DAPI staining, on the one hand, and fast bioinformatic evaluation tools based on it, such as flowCyBar [43] and flowEMMi v2 [37], on the other hand, a completely new online script for cell handling was developed.

The new procedure provides an online automated workflow with a high frequency of sampling (every 45 min) without the need for an operator, thereby eliminating human error. The generation of high-frequency and high-resolution datasets allows the determination of community states at the level of generation times, providing information on the evolution of a community. By using bioinformatics tools, such as flowCyBar, and correlation analyses between the cell subsets and reactor operational parameters [20], active or inactive members of a community can be accurately determined over particular time periods. The data processed by these tools can be provided by the automated gating tool flowEMMi v2 [37], which is also used in this study. Here, we offer an automated workflow that starts with automated sampling, dilution, fixation, staining and measurement with the OC-300 CytoFLEX instrumentation. The resulting fingerprints can also be analyzed in an automated manner using the flowEMMi v2 tool, allowing the resulting data to be fed into available bioinformatics tools. This makes the data available in a very short time, allowing for the bioreactors to be controlled online.

### 4.1. Development of an Automated Online Procedure to Obtain Fingerprints of Microbial Communities

The cell densities measured in this work ranged from 4.3 × 10^6^ to 4 × 10^9^ cells/mL. In principle, the CytoFLEX can measure cell densities down to 1 × 10^5^ cells/mL, while the OC-300 can dilute high cell densities up to 1 × 10^10^ cells/mL, together allowing the analysis of a wide range of cell densities. The cell density analysis is combined with DAPI staining, which, on the one hand, separates the cells from the instrumental noise and, on the other hand, creates fingerprints of the bacterial communities per sample point. The DAPI staining required a fixation step with NaCl, NaN_3_ and EtOH. NaCl increases the osmotic pressure, while EtOH creates pores for dye uptake by removing lipids from the cell membrane. NaN_3_ acts as a bacteriostatic that stops the metabolism of the cell and impedes the excretion of the dye by the cell [50,51]. The concentration of the DAPI solution was increased from a value of 0.24 µm, as described by Cichocki et al. (2020) [38], to 1 µm to achieve a higher fluorescence intensity of the cells for their better separation from the background noise of the CytoFLEX. The MC fingerprints generated by the OC-300 CytoFlex instrumentation were successfully verified by comparative measurements on the Influx (BD Biosciences, Franklin Lakes, NJ, USA) using both the standard and newly developed methods (Appendix A).

### 4.2. Reliable Self-Cleaning Procedures of the OC-300

The OC-300 automation unit, in combination with the CytoFLEX, uses many fluidic paths, e.g., tubing, chambers, syringes and flow cells, from one sample to the next. Therefore, a carryover between samples was to be expected, and we decreased it from 1% to 4% after testing several solutions (Table 1). A variation of 1% to 4% can be considered as being in the range of biological replicates of flow cytometric fingerprints (Appendix A).

As shown in Table 1, washing with only water was insufficient for properly cleaning the OC-300 between measurements. This could have been caused by bacterial cells adhering to the tube surfaces inside the unit. It has been shown that the strains of both *P. polymyxa* and *S. rhizophila* tend to adhere easily to different surfaces by means of adhesion proteins [52,53]. Adhesion mechanisms have also been suggested for *Kocuria* spp. [54]. For these reasons, it was obviously not enough to wash with water only. Subtilisin, a protease in the FlowClean cleaning agent, cleaves adhered cells from the surfaces and, in conjunction with the detergent ethoxylated nonylphenol, which is also present in the cleaning solution, helps to remove most of the remnants of the cells inside the tubes of the OC-300. The different levels of effectiveness of the cleaning solutions observed for the individual strains (Table 1) can also be explained by how susceptible each strain is to adhesion. We have found a balance between the amount of cleaning solution that can be used without affecting the bacterial patterns obtained by the automated workflow. As seen in Appendix A, the bacterial patterns are affected when using the 10% and 100% FlowClean cleaning agents. This is probably caused by remnant droplets of the cleaning solution after a two-step water rinsing procedure. These droplets may have a proteolytic effect on the bacteria, and this appears to have led to a change in the patterns seen in *K. rhizophila* after staining. The droplets potentially left behind by the 1% cleaning solution are obviously subcritical; therefore, we do not see the same effect on the bacterial patterns. To avoid a change in bacterial patterns and assure a proper cleaning between measurements, the use of the 1% FlowClean Cleaning agent is recommended.

### 4.3. Reliability of Automatically Determined Cell Counts

Figure 4a shows that the automated workflow is capable of providing stable cell concentration measurements over a period of at least 34 h without interruption due to an OC-300 or CytoFLEX malfunction. Only one major deviation in the automatic measurements was observed in the first four sampling points. Obviously, the OC-300 CytoFLEX instrumentation needs to stabilize for about 3 h before stable measurements can be performed.

Irrespective of the initial instrumental variations, we found variations in cell numbers caused by changes in the morphology of cells in the cell populations. In particular, *P. polymyxa* showed a marked increase in cell numbers over time. We hypothesize that this increase in cell concentration is not caused by bacterial growth or the variation introduced by the OC-300 but rather by aggregates of *P. polymyxa* breaking down into single cells caused by physical forces, i.e., stirring. Using the flowEMMi v2 software, two different subpopulations, C1, with only one chromosome content per cell and C2, with their doublets, were automatically differentiated, and the C1 subpopulation clearly increased in cell number in proportion to C2. The trend becomes even clearer when non-Gaussian distributions are included by setting an additional gate by hand via the FlowJo software that also includes the Cx subpopulation, which comprises triplets or more cells sticking together. This subpopulation decreases to two-tenths of its original abundance during 34 h of stirring, with cell abundances increasing, mainly in C1 but also in C2 at the beginning. The physiological state changes of the populations when sampling the identical sample over 34 h indicate that the type of sampling procedure is important for the pattern analysis. As long as the sampling is completed online, any changes in state can be attributed to the biotechnological process. However, if the sampled cells are collected and maintained in any solution for a longer time before measurement then changes in state can occur despite fixation. This shows that an online analysis is a particularly reliable workflow for monitoring and interpreting the biotechnological processes involving microbial communities.

## 5. Conclusions

Online automated flow cytometry can rapidly provide individual cell data that reflect the status of a bacterial community. A fast acquisition of information allows for highly time-resolved monitoring of a population’s dynamics and can be used as a basis for properly controlling the biotechnological processes that use bacterial communities. This is of particular relevance for the intended transition to a circular bioeconomy, which is based on the use of microbial consortia for the revalorization of waste.

Our online automated flow cytometry procedure is capable of providing the same high-resolution fingerprints of a bacterial community as a more tedious manual procedure. To achieve this high-resolution fingerprint, the bacterial sample first has to be diluted to a cell number measurable to the flow cytometer. After this dilution, the sample undergoes NaCl, NaN_3_ and EtOH fixation steps before the cells are stained with the DAPI dye.

The online automated flow cytometry procedure also provides stable cell counts and allows for monitoring bacterial communities at a temporal resolution similar to generation times. The automatic data acquisition can be coupled with the bioinformatics tool flowEMMi v2 for automatic gating as a prerequisite for the automatic evaluation of the data with other established bioinformatics tools. The resulting data can enable online control of the biotechnological processes involving microbial communities.

## Figures and Tables

**Figure 1 cells-12-01559-f001:**
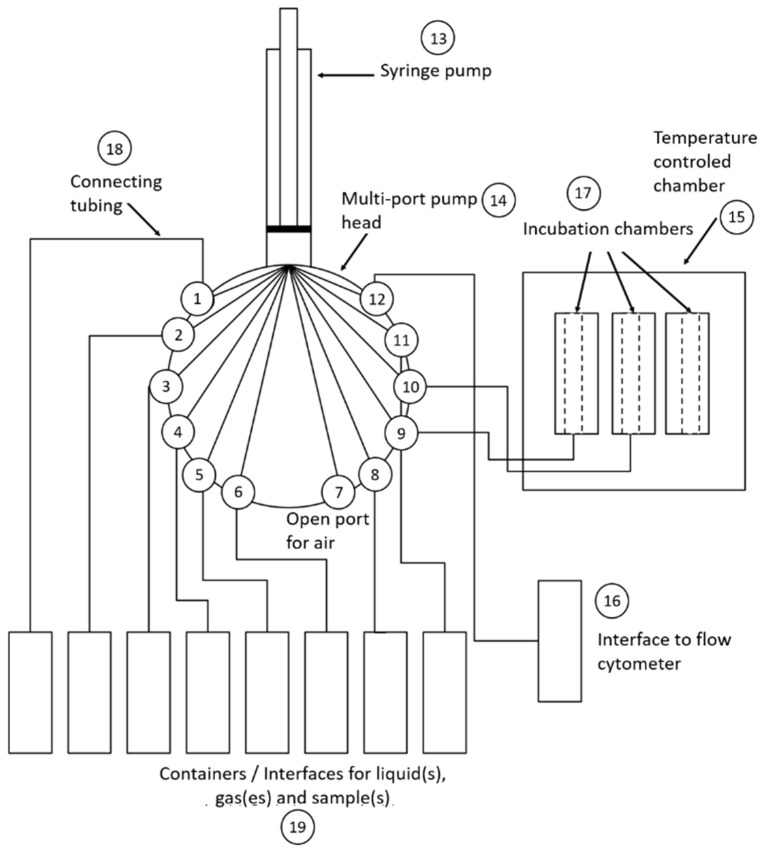
Scheme representing the components and connections of the OC-300 automation unit. (1–12) are the different ports for the connections to the different reagents, chambers and interface to the flow cytometer located in the multi-port pump head (14). (13) is the syringe pump. (15) is the temperature-controlled chamber with the incubation chambers (17) inside it. (16) is the interface that connects the OC-300 to the flow cytometer. (18) represents the tubing used for all the connections, and finally, (19) represents all the different flasks for the necessary reagents and solutions. Modified from Hammes and Weilenmann (2019) [42].

**Figure 2 cells-12-01559-f002:**
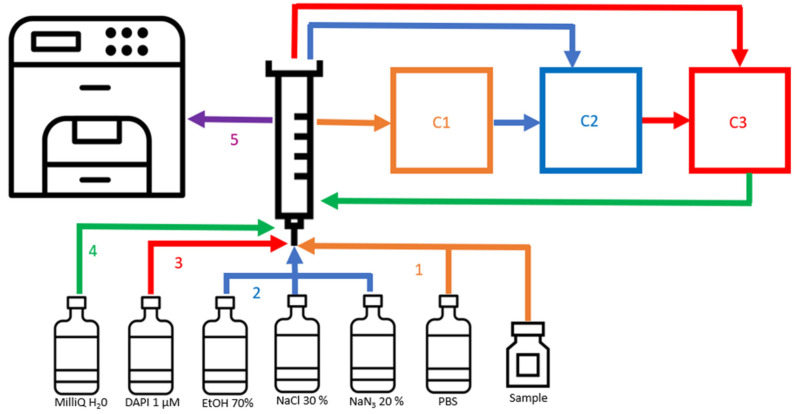
Scheme representing the workflow of automated dilution, fixation, staining and measurement. (1) The sample is diluted 1:1 with PBS and moved to chamber 1 (C1). (2) Then, the sample is moved from C1 to chamber 2 (C2) and fixed with NaCl 20%, NaN_3_ 1% and EtOH 10% for 10 min. (3) After, the fixed sample is moved from C2 to chamber 3 (C3) and stained with DAPI 1 µM for 10 min. (4) The stained sample is diluted 1:20 with MilliQ H_2_O in the syringe, and (5) the diluted sample is sent to the cytometer to be measured.

**Figure 3 cells-12-01559-f003:**
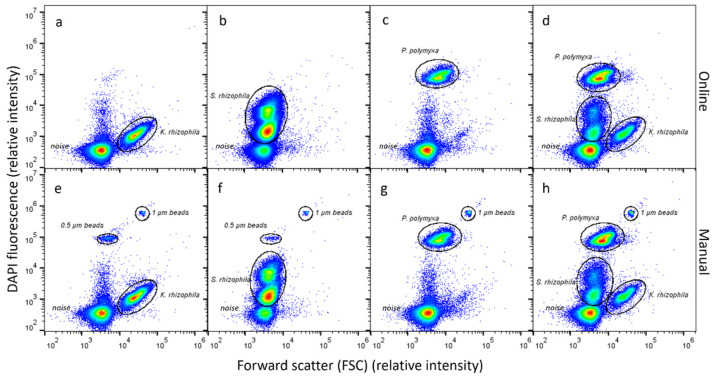
Comparison of the patterns of three pure strains, harvested in the stationary phase of growth, when performed automatically (*K. rhizophila*, *P. polymyxa*, *S. rhizophila*; (**a**–**c**) and manually (**e**–**g**). The microbial community MC is also both measured automatically (**d**) and manually (**h**). The *Y*-axis represents the measured DAPI fluorescence, and the *X*-axis the forward scatter. Further, 0.5 µm and 1.0 μm UV fluoresbrite microspheres were added to the manual measurements. Instrumental noise is clearly separated from the pure strains marked by gates and the MC under all conditions. The color marks the density of the cell abundance.

**Figure 4 cells-12-01559-f004:**
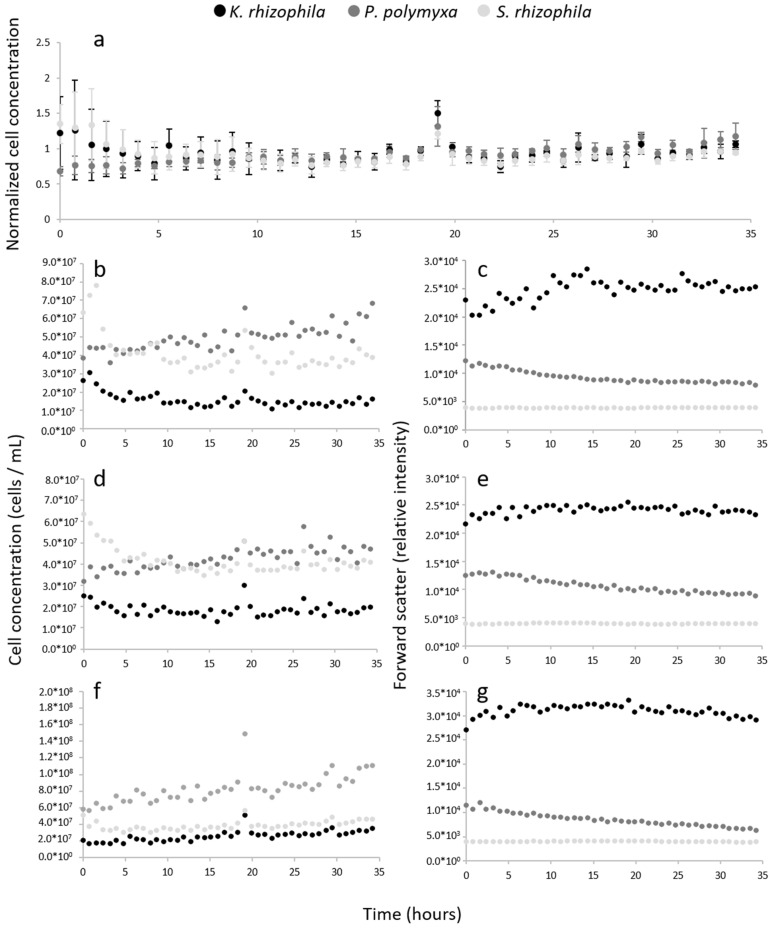
Pre-fixed PFA/EtOH cells of the MC were analyzed online using the OC-300 coupled to the CytoFLEX. Three biological parallels were performed after independently repeated cultivations of each of the strains. (**a**) Normalized cell concentrations of the three replicates over time and measured using the OC-300 automation device. (**b**,**d**,**f**) Cell concentration over time. (**c**,**e**,**g**) Forward relative scatter intensity changes over time.

**Figure 5 cells-12-01559-f005:**
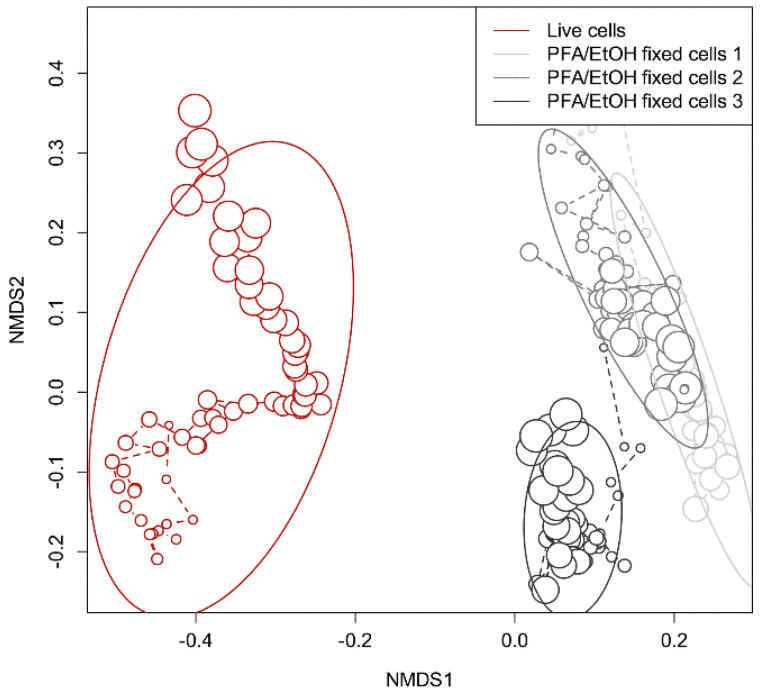
Divergence of the composition of the MCs, processed either from live cells or pre-fixed cells, measured and visualized by an NMDS plot: increasing size of circles represents an increasing time of sampling. Red shows the experiment conducted using live cells as input samples for the online OC-300 procedure. Different shades of gray represent the three experiments using PFA/EtOH pre-fixed cells (1–3) as input samples for the automated workflow. Ellipses represent the confidence region for each experiment with an alpha = 0.8.

**Figure 6 cells-12-01559-f006:**
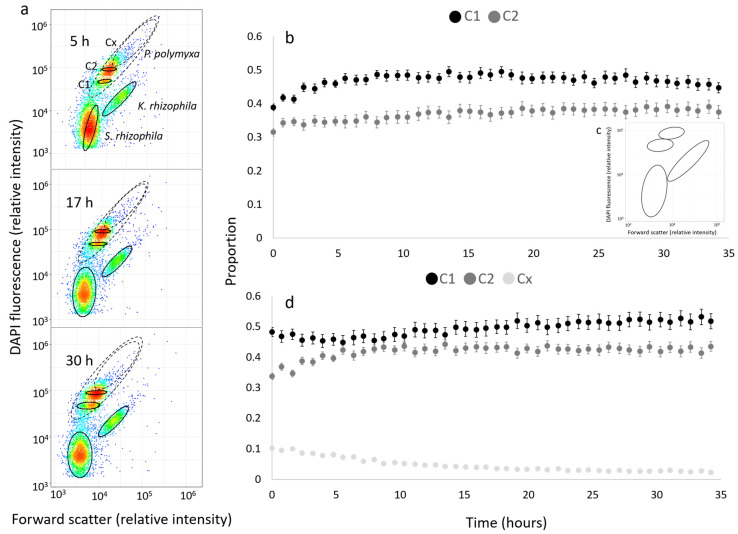
(**a**) Automated measurements of the MC at different time points which were independently evaluated by automatic gating (solid lines) using the flowEMMi v2 software. The dotted *P. polymyxa* and Cx gate were added manually for rapid identification of this strain. (**b**) Change in the proportions, from 0 to 1, of the subpopulations C1 and C2 with respect to the total cell count of the population of *P. polymyxa* over time. C1 represents the subpopulation comprising cells that contain only one chromosome content per cell, and C2 represents cells with two chromosome content per cell or doublets. Automated calculation of the cell numbers in the C1 and C2 subpopulations was done using the flowEMMi v2 software using the gate template shown in (**c**). Error bars show the standard error of the mean. (**c**) Automatic gate template generated by flowEMMi v2 for the whole online run after merging 132 individual automatic gatings of each sample. (**d**) Change in the proportions of the subpopulations C1, C2 and Cx of *P. polymyxa* over time. Calculation of the cell numbers in the C1, C2 and Cx subpopulations was completed using the FlowJo software. Error bars show the standard error of the mean. The color marks the density of the cell abundance.

**Table 1 cells-12-01559-t001:** Testing of the cleaning performance of four different cleaning solutions: MilliQ H_2_O, undiluted 100% FlowClean cleaning agent, 10% FlowClean cleaning agent, and 1% FlowClean cleaning agent using three different strains: *P. polymyxa*, *K. rhizophila* and *S. rhizophila*. Each strain was run alternately with the MilliQ H_2_O for a period of 24 h for every treatment; this means that 10 repeated measurements for the bacterial sample and 10 MilliQ H_2_O samples were performed. The carryover percentage represents the relation between cell concentration in the MilliQ H_2_O sample against the bacterial sample. Coefficients of variation are shown. All bacterial samples were prepared separately.

		MilliQ H_2_O	100% FlowClean Cleaning Agent	10% FlowClean Cleaning Agent	1% FlowClean Cleaning Agent
** *P. polymyxa* **	Cell concentration in bacterial sample (cells/mL)	5.14 × 10^8^ ± 20.1%	6.25 × 10^8^ ± 25.7%	7.02 × 10^8^ ± 25.6%	2.83 × 10^8^ ± 21.5%
	Cell concentration in water sample (cells/mL)	3.88 × 10^7^ ± 18.4%	4.89 × 10^7^ ± 29.7%	2.20 × 10^7^ ± 24.7%	1.08 × 10^7^ ± 11.7%
	**Carryover percentage**	**7.6 ± 2.3%**	**7.83 ± 2.7%**	**3.1 ± 0.4%**	**3.8 ± 1.0%**
** *K. rhizophila* **	Cell concentration in bacterial sample (cells/mL)	4.99 × 10^8^ ± 10.9%	2.65 × 10^8^ ± 10.0%	4.82 × 10^8^ ± 21.0%	4.12 × 10^8^ ± 11.4%
	Cell concentration in water sample (cells/mL)	1.54 × 10^7^ ± 23.4%	8.39 × 10^6^ ± 12.2%	1.02 × 10^7^ ± 21.6%	4.37 × 10^6^ ± 15.1%
	**Carryover percentage**	**3.1 ± 3.1%**	**3.2 ± 0.5%**	**2.1 ± 0.8%**	**1.1 ± 0.2%**
** *S. rhizophila* **	Cell concentration in bacterial sample (cells/mL)	1.25 × 10^9^ ± 21.2%	4.05 × 10^9^ ± 9.1%	1.51 × 10^9^ ± 8.37%	1.39 × 10^9^ ± 10.5%
	Cell concentration in water sample (cells/mL)	4.27 × 10^7^ ± 6.3%	9.66 × 10^7^ ± 7.9%	4.22 × 10^7^ ± 18.5%	3.85 × 10^7^ ± 12.4%
	**Carryover percentage**	**3.41 ± 0.6%**	**2.4 ± 0.2%**	**2.8 ± 0.5%**	**2.9 ± 0.6%**

## Data Availability

All additional data are available in the Appendix A. The raw datasets can be found in the Harvard Dataverse database.

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
