# Peer review of "Development of an Automated Online Flow Cytometry Method to Quantify Cell Density and Fingerprint Bacterial Communities"

_cells, 2023, doi:10.3390/cells12121559_

Round 1

Reviewer 1 Report

The article discusses interesting uses of the Flow cytometer, which we found useful. We would like to comment on a few points of interest.

Major comments:

More detailed information on the bacteria used would be helpful, such as the significance of staining with DAPI, distinguishing between genome size and FSC, and mentioning the shape and standard size of the bacteria, as different sized fluorescent beads are used, would also help to better understand the results in Figure 3.

The representation in Fig. 6 should be improved as it is difficult to understand. Is there three types of bacteria shown in the top panel of Fig. 6a, it should be clarified which gate P.polymyxa is shown in. Also, the top panel of Fig. 6a shows C1, C2 and Cx, but there is no indication of this in the middle and bottom panels, so it is unclear where Figs. 6b and 6d are tracing.

You may have mentioned this somewhere in the text, but it is worth reiterating: what does the noise component in the DAPI-FSC plot represent? Is it a bacterial-derived adulteration that has occurred during unstaining, manipulation or processing, or is it a chemical adulteration in the solution? If it is of bacterial origin, if the method is seen as a tool for tracing the composition and type of bacteria, does it require more analysis of the noise component?

Minnor comments:

There were several areas where we felt there were shortcomings in terms of representation. It is preferable to use the full spelling of bacterial names in the first publication and then abbreviate them (Lines 117-120)." It should be explained at the outset that the "SI" and the markings represent supplementary information.

The text inside Figures 1 and 2 is small and should be enlarged.

The notation of optical density should be standardised to either line 124 or line 151.

Author Response

Dear reviewer,

We have addressed all comments and questions from your side step by step and believe that the review process has significantly improved the quality of our manuscript. We would like to thank you for your contributions. Changes in the manuscript are highlight in grey.

Reviewer 2 Report

When reading the work, one gets a double impression. On the one hand, it is necessary to monitor the growth during the growth of the bacterial culture. The idea of bacterial fingerprinting in the analysis of bacterial communities is also attractive. On the other hand, the article looks like a flyer for the OC-300, but, unfortunately, after reading it, there are more questions than answers. At the moment, the article (and this work is presented as experimental) is not quite ready for submission to a peer-reviewed journal. There are many questions as to the setting of experiments, and, accordingly, the conclusions drawn based on the results. It is necessary to redo it, taking into account both the requirements of a particular journal, and bringing it into the classic form that readers and researchers are accustomed to.

1.      The introduction could have been made more lapidary, that is, concise and to the point. For example, lines 37-44 seem redundant.

2.      The phrase "functionally active subsets of cells" is not entirely clear in the context of this article, even in the introduction; you do not evaluate the functional activity of these pathogens in relation to their hosts using the toolkit you describe.

3.      The suggested cell density estimate will not solve the problem you describe in l.53-57, with biomass.

4.      L. 108-109. « This requires the cells to be automatically fixed and stained according to the specifications of a protocol by Günther et al.» According to a protocol Günther et al. «A mixture of 5 mM barium chloride and nickel chloride, each and 10% sodium azide was found to be a suitable fixative for all tested bacteria». You used the NaCl/Na-azide/EtOH fixation, could you please explain what caused this difference.

5.      L. 125-126. «was calculated (30°C, 24 h, 150 rpm) and used to inoculate 100 mL of LB medium» - this is not a good description of the technique. L.126. Before inoculation what final optical density OD 700nm do you have?

6.      L. 128-132. According to your reference (Cichocki et al. (2020)), authors proposed to used «the mock community consisted of two Gram-positive and two Gram-negative bacterial strains». Why then did you only use three cultures?

7.      L. 133. Please change ”NaCl/Na-azide/EtOH” to ”NaCl/NaN3/EtOH”

8.      L. 145 How many cells (what volume of cell suspension at such and such a density) did you use for fixation?

9.      L.151 “Fixed cells were washed with PBS buffer” it means that you have the step of washing, centrifugation, please, describe it.

10.  For such objects, what voltage did you give on the channels (FSC and SSC) during cytometry?

11.  It is not necessary to describe the equipment in detail in subsections 2.5 and 2.6. If interested, people can request information on the company's website.

12.  Please describes how the solutions in the cuvettes are mixed when adding, for example, DAPI? Because you always write that it is “online automated procedure”.

13.  L. 200. No need to describe the preparation of water, you can write in short - MilliQ H2O.

14.  «Figure 2. Scheme representing the workflow of automated dilution, fixation, staining and measurement». The scheme is strange - that is, after staining with DAPI, your sample is diluted with water and sent to the cytometer using the same syringe? What does mean the abbreviation “UPW”?

15.   L. 223- 241. It is not entirely clear what was done: that is, after work, a sample was taken from which tank? Of all or the last? How much technical iteration? Why the commercial detergent is used, rather than self-made and tested, so that researchers (readers of your article) can prepare themselves. The manufacturer recommends using a 100% solution and not diluting it, what is the rationale for your choice?

16.  Please, check “In addition, this software was also used for the analysis of subpopulations of P. polymyxa (C1, C2 and Cx), S. rhizophila and, K. rhizophila”, maybe more correct to write “P. polymyxa, S. rhizophila and, K. rhizophila (C1, C2 and Cx, respectively)”

17.  L. 276. “but also to distinguish subpopulations of some strains with different chromosome copy numbers”…. In order to isolate/distinguish bacteria with different sets of chromosomes, you must verify and validate your method. In this case, you know exactly how and what you mixed in your sample.

18.  L.259 – 263, L. 271 -281, L. 298-303 - These proposals are superfluous in the results section, they should be moved to the discussion.

19.  L. 282-288. Either you present optimization results or you need to remove this paragraph.

20.  Figure 3. Please explain – what means “noise” on the graph? It looks like a fourth bacterial population. Please describe in details what means – “performed online or manually”.

21.  Do I understand correctly that after washing with different solutions, you have 1-7 percents (Table 1) bacteria left in the sample? It is strange that it remains at all, it is necessary to select so that it is zero. Plus the second oddity - why does 100% FlowClean cleaning agent cope worse than its diluted counterpart?

22.  It should be better to rewrite the section 3.3. Reliability of automatically determined cell counts. Let the results be described briefly, but with numbers and timing (SI2.4). There are no calculations (L. 370-373), where you got these numbers from is not clear, but variations on the theme of 30% are not very good, but honestly.

23.  L. 416 – 418 How did you find and prove that this population consists of a C1 subpopulation, consisting of single cells with one set of chromosomes, and a C2 subpopulation, consisting of cells with a double content of chromosomes?

24.  Figures 5 and 6. What are “cells 3” and “Cx”on the graphs?

Author Response

(The authors gave the same response as above.)

Reviewer 3 Report

The main flaw of the presented research is the nature of analysed microbial cells which have been fixated prior to analysis. Although the measurement of cell density (cell concentration) is important from the point of view of many research and application aspects in the broadly defined microbiological analysis, it should be noted that in situ studies of microbial cells provide much more information about the complexity of the evaluated populations. It is primarily about determining the viability and metabolic activity of microorganisms which, in the face of the presence of microbial cells in dormant and damaged forms, are of great importance in the aspect of e.g. the assessment of microbiological safety in the food industry. The same will apply to the analysis of the potential for antibiotic resistance in clinical applications and broad aspects of microbes applied in biotechnological processes. In this case, the presence of micro-organisms from the group of so-called persisters can only be assessed by examining unfixed cells.

The presented results represent a significant step forward in modern microbiological analysis. They allow the assessment of the concentration of microbial cells in combination with the discrimination of types and even species, even in relation to bacterial cells, which due to their size, they are always more challenging in cytometric analysis. The designed procedure, demonstrating the effectiveness in distinguishing bacterial species, proved the significant potential of this type of determination. It is noteworthy that the cell staining protocol itself is actually very simple - while the authors were able to obtain a stable pattern of gating strategies through advanced optimization. The assay designed in this way is characterized by full repeatability and precision of determination.

Therefore, even if the use of the designed assay in the detection of VBNC forms in industrial practice (food industry) is somewhat limited (detection of these forms of microorganisms is highly prospective due to the threat posed by their presence in food matrices), it does not diminish the fact that the designed procedure shows high analytical value. Particularly noteworthy is the combination of a cytometric protocol with a set for automatic preparation and staining of samples. This allows not only the automation of the diagnostic protocol but, above all, real-time operation. In fact the Authors has developed complex and fully equipped system covering hardware, software, and an automated workflow to monitor both the absolute cell abundance and community composition online, in real time. This is definitely the future of modern diagnostic techniques.

Some specific comments and recommendations are listed below:

Lines 117-120: I would suggest to use the full names of 3 bacterial strains (Kocuria rhizophila, Stenotrophomonas rhizophila, and Paenibacillus polymyxa) in Materials and Methods section, prior to use it in the Results section.

Line 308: I wonder why the Authors used the stationary phase cultures of the bacterial strains? This will not always be the case in real diagnostic conditions as there will be a mixture of growth phases. It is however very difficult to simulate all growth conditions even if it affects to some extent the cellular morphology.

Line 423: I am not sure whether those suggested interconnected duplicates are the cells which have not yet undergone complete cell division or maybe they are naturally occurring cells duplicates similar to those cellular arrangements characteristic for gram-positive coccus shape bacterias, eg. diplococcus, streptococcus, tetrad, sarcina, etc. Or perhaps the fixation protocol itself leads to such double cell forms. This should be thoroughly checked by comparison with live cells from different growth stages.

Author Response

(The authors gave the same response as above.)

Reviewer 4 Report

The manuscript describes a method development study for the simultaneous measurement of bacterial strains by an online flow-cytometry instrument to monitor the bacterial populations. 

The study design, implementation, evaluation, and presentation are precise, however, some methodological questions were raised during the reading of the manuscript:

- FSC-DAPI dot plots were used in the measurements. Did you check FSC-SSC dot plots to identify and gate bacterial populations at first on FSC-SSC, then use the gated populations on FSC-DAPI dot plot to reduce noise and improve the selectivity?

- Did you set/apply a threshold on any of the channels? Could be the noise eliminated by setting the threshold? 

- In Fig 6a the noise seems to be eliminated. If it is, please explain (also in the text) how did you do it. 

-Please discuss the limitations of the study and method. It seems to be the application of a "clear" mixture/culture of the bacterial strains, but what is the situation if the culturing environment is a more complex biological environment in biotechnological applications (containing more debris, aggregates, and contaminants)? 

Only minor check is required.

Author Response

(The authors gave the same response as above.)

Round 2

Reviewer 2 Report

Unfortunately, I did not see a significant improvement in the presentation of the results (Results section), and the writing of the article as a whole (Introduction, Material and Methods sections), also there are no clear justifications / explanations for the results obtained - the presence of noise, the use of a commercially available diluted cleaning solution. There is no clear explanation, supported either by additional experiments or references to the literature.
